# Serum proteomic characterization of stump-tailed macaques (*Macaca arctoides*) with neutralizing antibodies against Dengue virus in Thailand

Pakorn Ruengket[1], Sittiruk Roytrakul[2], Daraka Tongthainan[3], Kobporn Boonnak[4], Kanokwan Taruyanon[5], Bencharong Sangkharak[6], Wirasak Fungfuang[7]*

1 Genetic Engineering and Bioinformatics Program, Graduate School, Kasetsart University, Bangkok, Thailand, 2 Functional Proteomics Technology Laboratory, National Center for Genetic Engineering and Biotechnology (BIOTEC), National Science and Technology Development Agency, Pathum Thani, Thailand, 3 Faculty of Veterinary Medicine, The Rajamangala University of Technology Tawan-ok, Chonburi, Thailand, 4 Department of Immunology, Faculty of Medicine, Siriraj Hospital, Mahidol University, Bangkok, Thailand, 5 Wildlife Conservation Division Protected Areas Regional Office 3, Department of National Parks, Wildlife and Plant Conservation, Ratchaburi, Thailand, 6 Wildlife Conservation Division, Department of National Parks, Wildlife and Plant Conservation, Bangkok, Thailand, 7 Department of Zoology, Faculty of Science, Kasetsart University, Bangkok, Thailand

* fsciwsf@ku.ac.th

## Abstract

Dengue infection is a leading global public health problem, and the virus (DENV) is transmitted by *Aedes* mosquitoes and causes a wide spectrum of clinical manifestations in humans. Previous studies have shown that wild macaques in Thailand have been exposed to the dengue virus. Accordingly, this study aims to characterize the serum proteomic profiles of wild macaques with (seropositive) and without (naive) dengue virus-neutralizing antibodies, to improve our understanding of differential protein expression and identify candidate biomarkers. We analyzed thirty-two serum samples obtained from wild macaques in Thailand. Out of the 32 samples, 16 macaques (50%) were seropositive for DENV. A total of 9,532 proteins were identified, out of which 582 were differentially expressed (DEPs) and used to generate a proteomic profile. Among the nine identified proteins of interest, three were associated with the nervous system, while the remaining six have been reported to play roles in anti-dengue antiviral mechanisms. These include the induction of interferon responses triggered by cellular stress, degradation of viral RNA, and activation of dendritic cells and Th1-mediated immune responses via IFN-γ and TNF-α. Although these proteins have been implicated in the nervous system and are involved in anti-DENV in several reports, in this study, it was shown that these proteins remain up-regulated even after infection, which requires further study into the dynamics after long-term infection in the future. These findings highlight the need for further longitudinal studies to validate the functional relevance of these proteins.

**Data availability statement:** The datasets generated and/or analysed during the current study are available in the jPOST, PXD052111 (https://proteomecentral.proteomexchange.org/cgi/GetDataset?ID=PXD052111) and the data sets supporting the conclusions of this article are included with the article and its additional files.

**Funding:** This research was supported by the Thailand Research Fund, Thailand (Grant No. MRG 6080260). The funders had no role in the study design, data collection and analysis, the decision to publish, or the preparation of the manuscript.

**Competing interests:** The authors have declared that no competing interests exist.

**Abbreviations:** ADE: Antibody-dependent enhancement; BFGS: Broydene-Fletcher-Goldfarbe-Shanno; CID: Collision-induced-dissociation; CDK4: Cyclin-dependent kinase 4; Cysteine CAM: Cysteine carbamidomethylation; DHF: Dengue hemorrhagic fever; DSS: Dengue shock syndrome; DENV: Dengue virus; DEPs: Differentially expressed proteins; EDTA: Ethylenediaminetetraacetic acid; KOGs: EuKaryotic Orthologous Groups; GBP3: GB1/RHD3-type G domain-containing protein; GRIN2C: Glutamate ionotropic receptor NMDA type subunit 2C; HSPA4: Heat shock 70 kDa protein 4; HCA: Hierarchical clustering analysis; KO: KEGG Orthologous; GhostKOALA: KEGG Orthology, Links Annotation; KNN: K-nearest neighbors; LC-MS/MS: Liquid chromatography-tandem mass spectrometry; LR: Logistic regression; DF: Mild dengue fever; MBP: Myelin basic protein; SIRT4: NAD-dependent protein deacetylase sirtuin-4; NMDA: N-methyl-D-aspartate; PCA: Principal component analysis; PANTHER: Protein Analysis Through Evolutionary Relationships; PELs: Protein expression levels; PRNT90: 90% plaque reduction neutralization assays; RF: Random forest; RNASEL: Ribonuclease L; SVM: Support vector machine; TARS2: Threonyl-tRNA synthetase; UCHL1: Ubiquitin carboxyl-terminal hydrolase; WHO: World Health Organization.

## Introduction

Dengue virus (DENV) is a single positive-stranded RNA virus belonging to the genus *Flavivirus* and family Flaviviridae, and has four serotypes: DENV1–4 [1–3]. DENV is the causative agent of dengue fever, which is transmitted through the bite of infected *Aedes aegypti* and *A. albopictus* mosquitoes [4,5]. Although DENV infection is typically self-limiting, it can cause a wide spectrum of clinical manifestations in humans, ranging from asymptomatic or mild dengue fever (DF) to life-threatening dengue hemorrhagic fever (DHF) and dengue shock syndrome (DSS) [2,6]. A previous study reported that infection with a given serotype may confer lifelong immunity against homologous strains, while only short-term cross-protection is provided against heterologous serotypes [6,7]. However, reinfection with a different serotype can lead to severe dengue due to antibody-dependent enhancement (ADE) [6]. Furthermore, cytokines are key regulators in dengue pathogenesis, mediating the complex interactions between immune dysregulation and the progression of infection [8]. Recently, DENV has become the most widespread arthropod-borne viral disease and poses a major global public health concern, particularly in tropical and subtropical regions [9]. The World Health Organization (WHO) reported a nearly tenfold increase in global dengue incidence, from 505,430 cases in 2000 to 5.2 million in 2019 [10]. Modeling studies estimate that from the 390 million annual DENV infections, approximately 96 million result in clinical symptoms. DENV is now endemic in over 100 countries, with Southeast Asia accounting for approximately 70% of all reported cases [11,12].

WHO guidelines emphasize mosquito population control and a reduction in the potential breeding habitats as strategies to reduce and prevent dengue transmission. However, these methods have limitations: while effective in the short term, their long-term usage is financially challenging. The global economic burden of dengue is substantial, with annual costs estimated at US$8.9 billion [13,14]. Therefore, there is a pressing need for innovative and sustainable strategies to combat mosquito-borne diseases that have both long-term effectiveness and economic feasibility [14].

Moreover, DENV circulates through both a sylvatic (enzootic) cycle involving non-human primates and various *Aedes* mosquito species, and a human transmission cycle, primarily vectored by *A. aegypti* [15,16]. In Africa, the only sylvatic DENV serotype isolated to date is DENV-2. In contrast, sylvatic DENV-1, DENV-2, and DENV-4 have been isolated in Asia [16], although the last confirmed isolation of a sylvatic strain (DENV-4) occurred in 1975. Additionally, the so-called sylvatic isolate of DENV-1 is not phylogenetically distinct from human-derived lineages, leading to uncertainty about its origins [17]. Phylogenetic analyses suggest that sylvatic DENVs are ancestral to the strains currently endemic in human populations. Numerous non-human primate species, including chimpanzees, rhesus and cynomolgus macaques, sooty mangabeys, African green monkeys, Guinea baboons, common marmosets, and owl monkeys, have been reported to host DENV [4,18,19]. The extensive biodiversity and broad geographic distribution of macaques, from North Africa to Asia, indicate their importance as potential reservoirs of zoonotic diseases. Members of the *Macaca* genus are highly adaptable, capable of thriving in diverse habitats, which increases their likelihood of interactions with humans and other

animals. This ecological flexibility enhances their potential role in cross-species disease transmission. Previous studies have shown that rhesus macaques (*Macaca mulatta*) as one of the earliest non-human primate species experimentally infected with dengue virus. Despite their extensive use as an experimental model, these macaques seldom develop the hallmark clinical features of dengue fever [20,21]. Likewise, cynomolgus macaques (*Macaca fascicularis*) typically exhibit only transient clinical manifestations and low viremia, accompanied by antibody response profiles resembling those observed in human infections, yet without overt signs of disease [20,22]. In addition, the macaques exhibited no observable clinical symptoms, and no significant alterations were detected in body temperature, body weight, hematocrit levels, platelet counts, or red and white blood cell counts [23]. Pre-existing immunity to one of the four DENV serotypes is recognized to elevate the risk of severe disease during secondary infection with a distinct serotype. Similar to humans, macaques can experience Antibody-Dependent Enhancement (ADE), wherein pre-existing antibodies from a prior infection with one dengue serotype may intensify disease severity following infection with a heterologous serotype [23]. As such, understanding the role of macaques in zoonotic pathogen dynamics is critical for both public health planning and wildlife conservation efforts [24].

In Thailand, macaques inhabit a wide range of ecosystems, from forests to urban environments. However, their populations are increasingly threatened by habitat loss driven by deforestation, agricultural expansion, and urban development. These pressures not only reduce the available habitat but also increase the chances of human-macaque interactions, often resulting in conflict. In agricultural areas, macaques are known to raid crops, creating tension with local farmers. In urban settings, they may scavenge for food and are often regarded as nuisance animals. Such interactions further complicate efforts to manage disease risk and conserve primate populations [25,26]. Tongthainan D, et al. [27] identified the presence of neutralizing antibodies (seropositive) against DENV, Zika, and Chikungunya viruses in three distinct species of wild macaques native to Thailand, emphasizing the potential risk of viral disease transmission from animals to humans. The detection of neutralizing antibodies against DENV in serum indicates prior exposure of macaques to the virus, suggesting that the proteins identified in seropositive individuals may be associated with past immune responses. Several studies have shown that serum/plasma is a common biomarker for diagnosis and prognosis in a variety of diseases, as tissue pathology often leads to the release of certain proteins into the bloodstream [28,29]. Accordingly, this study employed proteomic approaches to investigate proteins potentially involved in the recovery from previous infection, long-term immune protection, or indicative of residual effects from past DENV-induced pathology and develop candidate biomarkers by comparing macaques with neutralizing antibodies against DENV to seronegative (naive) individuals.

## Materials and methods

### Ethics approval and consent to participate

All animal care and handling procedures were approved by the Institutional Animal Care and Use Committee of the Kasetsart University Research and Development Institute at Kasetsart University in Thailand (ID: ACKU67-SCI-006) in accordance with the National Institutes of Health's (U.S.A.) Guide for the Care and Use of Laboratory Animals. The experimental protocols for the study in a conservation area, obtaining of blood samples, and the release of wild macaques were approved by the Department of National Parks, Wildlife and Plant Conservation in Thailand (Permit Number: 0909.204/14187).

### Sample collection

Thirty-two blood samples were collected from 24 males and 8 females wild stump-tailed macaques (*Macaca arctoides*) living freely around the Pa La-U Waterfall in Huahin District, Prachuap Kiri Khan Province (GPS: 12.240800, 99.464004). 31 adult and 1 juvenile macaques were captured using baited cages (4 x 4 x 3; W x L x H), containing seasonal fruits. To minimize stress, the captured macaques were subsequently transported to a temporary veterinary station with opaque cloths during handling. All macaques were anesthetized via an intramuscular injection of tiletamine-zolazepam (2–5 mg/kg) and

xylazine hydrochloride (0.5–2 mg/kg). Blood was drawn from the femoral vein and collected in ethylenediaminetetraacetic acid (EDTA) tubes, with a maximum volume of 3 mL per tube. Samples were centrifuged at 2,200 × g for 20 minutes at 4°C, and the resulting plasma was stored at –80°C until further analysis. Following sample collection, each macaque was monitored during recovery in a cage that was placed in a quiet area. Respiratory rate, heart rate, and body temperature were continuously monitored during the anesthetic procedure. The recovery was defined as the return of normal posture and mobility, which typically occurred within 60–90 minutes post-anesthesia. After full recovery from anesthesia, all macaques were released back into their habitat.

## Serology assessment

Neutralizing antibodies to DENV1–4 were identified using the 90% plaque reduction neutralization test (PRNT90), based on the lowest serum dilution resulting in a 90% reduction in viral foci. Test sera were sequentially diluted by 4-fold starting at 1:5 using OptiMEM (Invitrogen, USA) containing 0.3% human serum albumin, which was heat-inactivated at 56°C for 30 minutes. DENV1–4 viruses were diluted to a final concentration of 1,000 pfu/mL and mixed in an equal volume with diluted serum, followed by incubation at 37°C for 30 minutes. Vero cell monolayers grown in 24-well plates (90% confluence) were infected by adding 50 µL of the virus-serum mixture to duplicate wells. After 60 minutes of adsorption at 37°C, cells were overlaid with 0.5% methylcellulose in OptiMEM supplemented with 2% FBS and incubated for 4 days. Plaques were visualized using immunoperoxidase staining. After removing the overlay, cells were fixed with 80% methanol for 30 minutes and blocked with 5% nonfat milk in PBS. Monoclonal antibodies 2H2 (anti-DENV1–4 prM) and 4G2 (anti-Flavivirus envelope E protein) were added at a 1:2,000 dilution and incubated for 1 hour at room temperature. After washing, peroxidase-labeled goat anti-mouse IgG (1:2,000) was added and incubated for 1 hour at 37°C. The plates were washed again, and plaques were developed using 4-chloro-1-naphthol in H2O2. Visible plaques were counted, and PRNT90 titers were calculated using the NIH/NIAID online tool (available at https://bioinformatics.niaid.nih.gov/plaquer-eduction). According to the WHO criteria for dengue seropositivity, a PRNT90 titer of ≥20 is considered seropositive for DENV [30–32]. According to these criteria, ZIKV infection is classified in samples with PRNT90 titer values greater than 20 and a 4-fold difference between ZIKV and DENV PRNT90 titers.

## Serum protein preparation

Using bovine serum albumin as the protein standard, the protein content of each serum sample was determined in accordance with the Lowry protein assay procedure [33]. Briefly, 5 mM dithiothreitol in 10 mM ammonium bicarbonate was added to 5 µg of serum, and the mixture was incubated at 60°C for 1 hour to reduce disulfide bonds. Next, sulfhydryl group alkylation, with 15 mM iodoacetamide in 10 mM ammonium bicarbonate, was carried out at room temperature for 45 minutes in the dark. The carbamidomethylated protein samples were then digested with sequencing-grade trypsin at a 1:20 ratio and incubated at 37°C overnight. The tryptic peptides were dried using a speed vacuum concentrator, and the resulting pellet was resuspended in 0.1% formic acid for nano-liquid chromatography-tandem mass spectrometry (LC-MS/MS) analysis.

## LC-MS/MS analysis

Tryptic peptide samples were injected into an Ultimate3000 Nano/Capillary LC System (Thermo Scientific, UK) coupled with an HCTUltra LC-MS system (Bruker Daltonics Ltd., Hamburg, Germany) using a Nano-captive spray ion source. Briefly, 5 µg peptide digests were enriched on a Precolumn 300 µm ID × 5 mm C18 PepMap 100, 5 µm, 100 Å (Thermo Scientific, UK), and separated on a 75 µm ID × 15 cm column packed with Acclaim PepMap RSLC C18, 2 µm, 100 Å, nanoViper (Thermo Scientific, UK). A column oven set to 60°C was used to house the C18 column. The analytical column was supplied with solvents A and B, which contained 0.1% formic acid in water and 0.1% formic acid in 80% acetonitrile, respectively. The peptides were eluted over 30 minutes using a solvent B gradient of 5–55% at a constant flow rate of

0.30 µL/min. Utilizing CaptiveSpray, electrospray ionization was carried out at 1.6 kV. Approximately 50 L/h of nitrogen was used as the drying gas. Collision-induced dissociation (CID) product ion mass spectra were captured using nitrogen as the collision gas. Positive-ion mode MS and MS/MS spectra were collected at 2 Hz, spanning a range of 150–2200 m/z. Collision energy was set to 10 eV based on the m/z values, and each sample was analyzed in triplicates through the LC-MS.

## Protein identification and quantification

LC-MS/MS data were quantified using DeCyder MS Differential Analysis Software, and the UniProt Macaca protein database was searched using the Mascot search engine to identify and annotate the peptide sequences generated from LC-MS/MS analysis [34,35]. Trypsin served as the digesting enzyme, and the following Mascot standard settings were applied: a maximum of three missed tryptic cleavages, a fragment peptide mass tolerance of 1.2 Da, an MS/MS tolerance of 0.6 Da, cysteine carbamidomethylation (Cysteine CAM) as the fixed modification, methionine oxidation as the variable modification, and peptide charge states of 1+, 2+, and 3+. Protein expression levels (PELs) from the MS/MS spectra were expressed as log2-transformed values.

## Statistical analysis

Neutralizing antibodies to DENV1, DENV2, DENV3, and DENV4 were detected using PRNT90 assays. Half of the samples were seropositive, while the remaining half were naïve. PELs were analyzed using the Mann–Whitney U test in IBM SPSS Statistics version 22.0.0 (Armonk, NY, USA) to compare total protein levels between the two groups at a significance level of $P < 0.05$. Fold change thresholds of 1.2 and 1.5 were applied to identify enriched protein subsets.

## Machine learning analysis

Four classification algorithms, which included random forest (RF), support vector machine (SVM), k-nearest neighbors (KNN), and logistic regression (LR), were used to generate classification prediction scores [36]. RF, structured as an ensemble of decision trees, minimized Gini impurity with a maximum tree depth of four and used 1,000 estimators. SVM employed a polynomial kernel of degree two to analyze data for classification [36,37]. KNN, a non-parametric, instance-based learning algorithm, was implemented with three nearest neighbours (K = 3) using Euclidean distance as the metric. LR modeled the binary output using a logistic function, with unconstrained nonlinear optimization performed iteratively via the Broyden-Fletcher-Goldfarb-Shanno (BFGS) algorithm. Leave-one-out cross-validation was used to determine the classification accuracy for each sample.

Unsupervised machine learning was applied to cluster the sample groups. Principal component analysis (PCA) was performed on the protein expression level (PEL) data to visualize protein distribution and clustering in a scatter plot. The 32 samples were grouped into two main clusters as either non-infected or infected, and further analyzed using hierarchical clustering analysis (HCA) based on PELs

## Bioinformatic data analysis

Gene orthology and functional categorization analyses were conducted by comparing proteins with a 1.5 FC to the EuKaryotic Orthologous Groups (KOGs) and KEGG Orthology (KO) databases. Functional protein association networks were constructed using the STRING database version 11.5 with a minimum interaction confidence score of 0.400 and query proteins only. Protein–drug–chemical interaction networks were generated using the STITCH database v5.0 with a high confidence threshold. Cytoscape version 3.9.0 was used to visualize networks from STRING and STITCH. Potential protein function was assigned by K number to the protein sequence using the Protein Analysis Through Evolutionary Relationships or PANTHER classification system, Reactome, KEGG Orthology, Links Annotation (GhostKOALA) databases, and HMMER, with cut-off E-value set to 1E–05.

## Results

### Neutralizing antibody against DENV

The PRNT90 assay was used to detect neutralizing antibody titers against DENV1, DENV2, DENV3, and DENV4 in serum. A PRNT90 titer >20 was considered DENV seropositive. All four DENV serotypes were neutralized in the serum samples. Of the 32 serum samples, 16 (50%) were seropositive for DENV antibodies. Overall, DENV-2 (36.84%) was the most prevalent serotype, followed by DENV-1 (31.58%), DENV-3 (23.69%), and DENV-4 (7.89%). Among the seropositive macaques, 15.63% showed monotypic and 3-serotype responses, while 9.38% showed 2-serotype and 4-serotype responses (Tables 1 and 2).

### Differential protein identification between two macaque groups

Samples from both naïve and seropositive macaques were analyzed in triplicate to reduce random sampling effects and improve the validity of protein detection. Log2 fold changes between paired values were calculated to identify proteins with a significantly altered expression due to infection. A total of 9,532 proteins were identified through shotgun proteomic analysis. Among these, 68 and 70 proteins were uniquely expressed in naïve and seropositive macaques, respectively, while 9,394 proteins were identified in both groups (Fig 1). Statistical analysis refined the protein dataset, isolating 582 differentially expressed proteins (DEPs), with 560 shared between groups and 13 and 9 uniquely expressed in naïve and seropositive macaques, respectively. To enhance the proteomic profile, fold-change thresholds of 1.2 and 1.5 were used, respectively. At 1.2-fold change, 94 proteins were upregulated and 488 downregulated, while the 1.5-fold change threshold identified 86 upregulated proteins.

The expression data were subjected to feature selection and supervised classification analysis to identify the subset of proteins that discriminated between naïve and seropositive macaques. A combination of three filter-based methods was used to select proteins with differential expression profiles between the two groups. From the original set of 9,532 proteins, using 1.2-fold and 1.5-fold upregulation thresholds resulted in subsets of 94 and 86 proteins, respectively (Table 3). Four classification algorithms, RF, SVM, KNN, and LR, were used for model training. The best classification performance was achieved with an 86-protein subset (1.5-fold change), with classification rates ranging from 93.75% (RF and KNN) to 100% (SVM and LR).

Subsequently, an unsupervised PCA was used to visualize the 86 feature proteins. The first two principal components accounted for 22.00% and 6.01% of the total variance, respectively (Fig 2). A clear separation between the naïve and seropositive groups was obtained using unsupervised hierarchical clustering of the 86 proteins (Fig 3).

### Construction of interaction networks

Functional classification of the 86 proteins was performed using the KO and KOGs databases. The obtained classification and categorization rates by KO and KOGs were 68.60% and 63.95%, respectively. However, combining both KO and KOGs analyses increased the classification rate to 72.09% (Fig 4A). Among these, 38.37% of the proteome was enriched in cellular processes and signaling, followed by information storage (22.09%), metabolism (11.63%), and unknown function (27.91%) (Fig 4B). Gene ontology enrichment analysis based on Molecular Function indicated that most proteins were involved in catalytic activity (26%), followed by binding (21%), transporter activity (5%), transcription regulator activity (3%), ATP-dependent activity (2%), molecular function regulator (2%), molecular transducer activity (1%), and structural molecular activity (1%) (Fig 4C). The biological processes included cellular process (26%), metabolic process (17%), biological regulation (11%), response to stimulus (5%), signaling (5%), localization (4%), multicellular organismal process (2%), reproduction (2%), reproductive process (2%), biological adhesion (1%), developmental process (1%), and immune system process (1%) (Fig 4D). Cellular Components included cellular anatomical entity (46%) and protein-containing complex (15%) (Fig 4E).

**Table 1. Neutralizing antibody titers against DENV in macaque serum.**

| Samples | PRNT90 | | | |
|---|---|---|---|---|
| | DENV1 | DENV2 | DENV3 | DENV4 |
| N01 | – | – | – | – |
| N02 | – | – | – | – |
| N03 | – | – | – | – |
| N04 | – | – | – | – |
| N05 | – | – | – | – |
| N06 | – | – | – | – |
| N07 | – | – | – | – |
| N08 | – | – | – | – |
| N09 | – | – | – | – |
| N10 | – | – | – | – |
| N11 | – | – | – | – |
| N12 | – | – | – | – |
| N13 | – | – | – | – |
| N14 | – | – | – | – |
| N15 | – | – | – | – |
| N16 | – | – | – | – |
| I01 | 63.5 | 280 | – | – |
| I02 | 22 | – | – | – |
| I03 | 70.4 | – | – | – |
| I04 | 69.7 | 92.3 | – | – |
| I05 | – | 97.6 | – | – |
| I06 | – | 70.1 | – | – |
| I07 | – | 364.2 | – | – |
| I08 | 1387 | 320.5 | 344 | 35.7 |
| I09 | – | 70.1 | 285 | – |
| I10 | 63.3 | 96.6 | 92.8 | – |
| I11 | 20.8 | 28.6 | 33.6 | – |
| I12 | 2278 | 281.4 | 429.9 | 33 |
| I13 | 138.1 | 80.9 | 30.6 | – |
| I14 | 2278 | 285.8 | 385.1 | 84.3 |
| I15 | 97.2 | 80.8 | 82.5 | – |
| I16 | 136 | 389.7 | 285.7 | – |

The initial serum dilution in this assay was 1:5, and seropositivity was defined as a PRNT 90 > 1:20 based on WHO criteria.

A protein–protein interaction network was constructed using STRING (Fig 5). The network formed one sub-network, with Heat Shock 70 kDa Protein 4 (HSPA4) identified as the hub protein. Additionally, eight other proteins were associated with inflammation, host defense, apoptosis, and immune response. These included Glutamate Ionotropic Receptor NMDA Type Subunit 2C (GRIN2C), Ubiquitin Carboxyl-Terminal Hydrolase (UCHL1), Myelin Basic Protein (MBP), GB1/RHD3-Type G Domain-Containing Protein (GBP3), Ribonuclease L (RNASEL), Cyclin-Dependent Kinase 4 (CDK4), NAD-Dependent Protein Deacetylase Sirtuin-4 (SIRT4), and Threonyl-tRNA Synthetase (TARS2) (Fig 6, Table 4).

**Table 2. Number of seropositive samples.**

| Neutralizing antibody against dengue serotype | Number |
|---|---|
| Naïve | 16 (50.00%) |
| Monotypic | |
| DENV-1 | 2 (6.25%) |
| DENV-2 | 3 (9.38%) |
| DENV-3 | 0 |
| DENV-4 | 0 |
| **Total monotypic** | **5 (15.63%)** |
| Multipletypic | |
| 2-serotype | |
| DENV-1/DENV-2 | 2 (6.25%) |
| DENV-1/DENV-3 | 0 |
| DENV-1/DENV-4 | 0 |
| DENV-2/DENV-3 | 1 (3.13%) |
| DENV-3/DENV-4 | 0 |
| *Subtotal 2-serotype* | 3 (9.38%) |
| 3-serotype | |
| DENV-1/2/3 | 5 (15.63%) |
| DENV-1/2/4 | 0 |
| DENV-1/3/4 | 0 |
| DENV-2/3/4 | 0 |
| *Subtotal 3-serotype* | 5 (15.63%) |
| 4-serotype | |
| DENV-1/2/3/4 | 3 (9.38%) |
| **Total mutipletypic** | **11 (34.38%)** |
| **Total** | 32 (100%) |

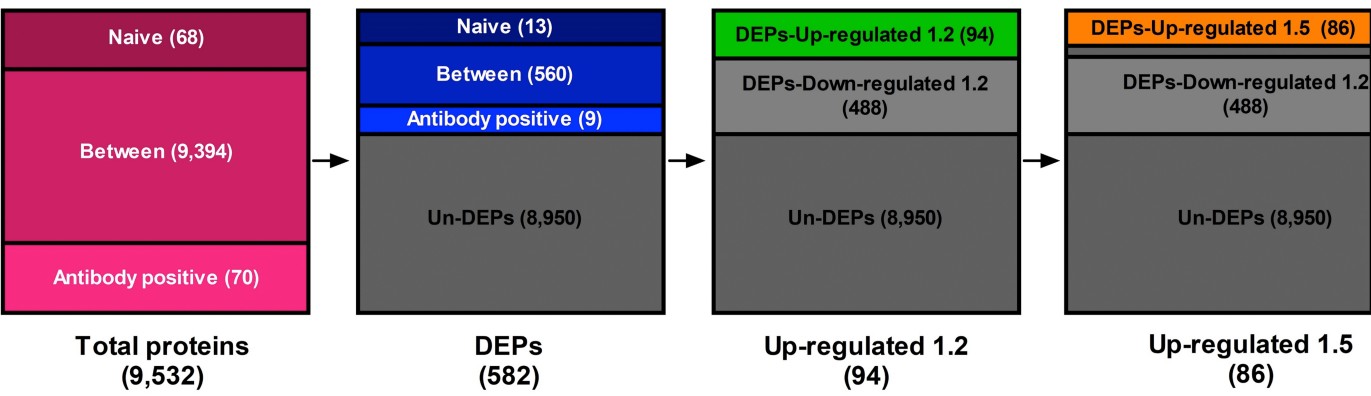

**Fig 1. Descriptive data.** The flow chart of filtration methods for the proteome profile.

## Discussion

The present study aimed to profile the differential serum protein expression in wild stump-tailed macaques with and without DENV-neutralizing antibodies. Our results showed that 50% of the macaques were seropositive for DENV. LC-MS/

**Table 3. Classification algorithm analysis.**

| Number of features | Total proteins (9,532 Proteins) | 1.2-fold-change up-regulated (94 Proteins) | 1.5-fold-change up-regulated (86 Proteins) |
|---|---|---|---|
| Classification algorithm | Accuracy | Accuracy | Accuracy |
| Random forest | 50.00% | 90.62% | 93.75%* |
| Support vector machine | 65.62% | 100.00%* | 100.00%* |
| K-nearest neighbors | 68.75% | 90.62% | 93.75%* |
| Logistic regression classifier | 65.62% | 96.88% | 100.00%* |

*Highest accuracy score.

**Fig 2. Unsupervised principal component analysis (PCA) based on 86 up-regulated proteins.** The PCA represents the summarized dimension with a total variance of 28.01% to distinguish between naïve and seropositive groups. The first and second dimensions account for 22.00% and 6.01% of the variance, respectively.

MS analysis of 32 serum samples identified a total of 9,532 proteins. To identify serum proteins with potential utility as biomarkers for detecting dengue-seropositive macaques, we prioritized molecules exhibiting upregulation proteins, as these are more amenable to reliable detection. Accordingly, we implemented a filter-based analytical framework integrating multiple bioinformatic approaches, including statistical testing, fold-change assessment, and both supervised and unsupervised machine-learning algorithms. Using a filter-based method, we initially identified 86 proteins that met

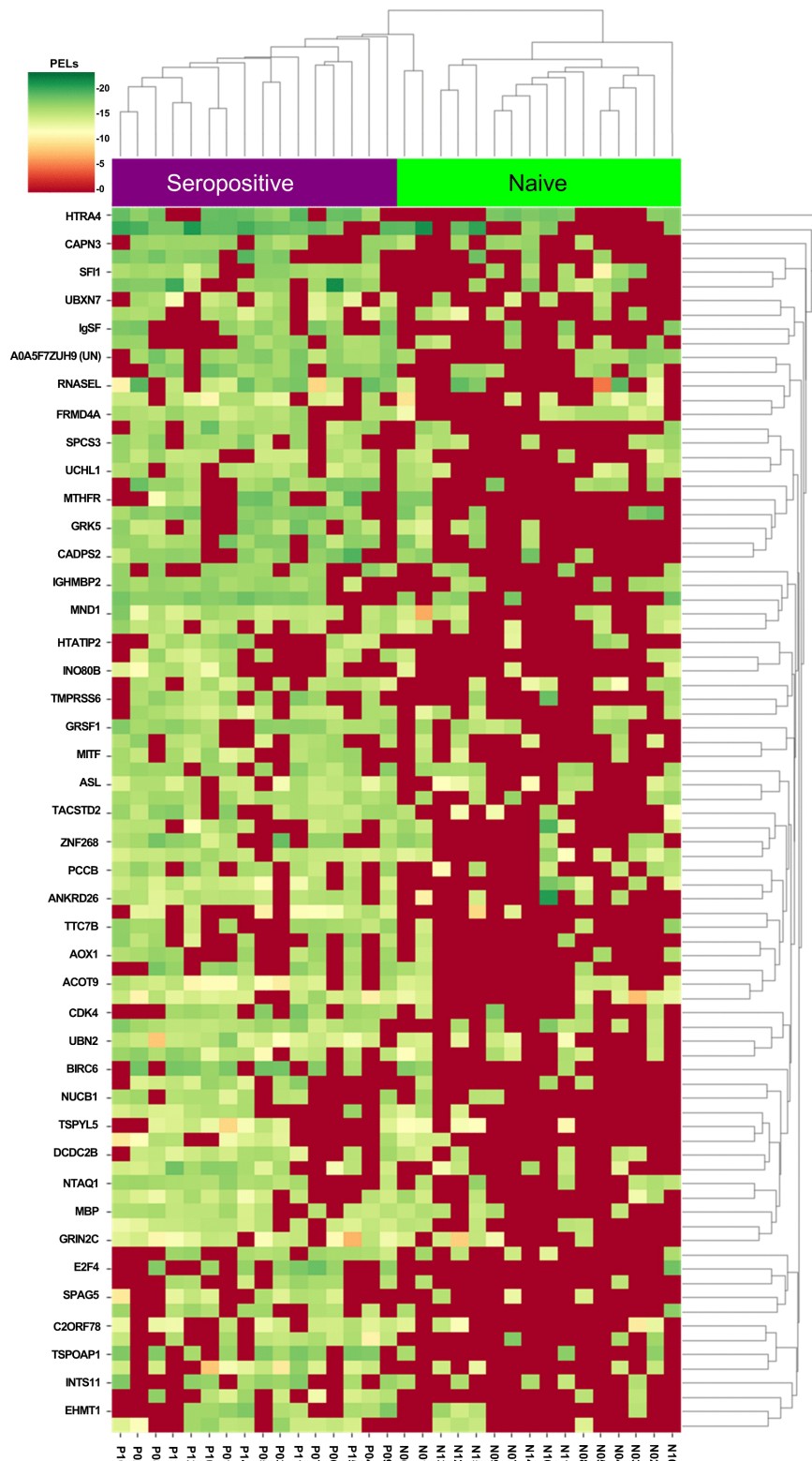

**Fig 3. Unsupervised hierarchical cluster analysis based on 86 up-regulated proteins.** Purple and green bands in the horizontal columns represent samples in the seropositive and naïve groups, respectively. The vertical rows indicate the 86 up-regulated protein names. The color of the heatmap indicates the scale of PELs with low-to-high expression in 32 samples and 86 up-regulated protein features.

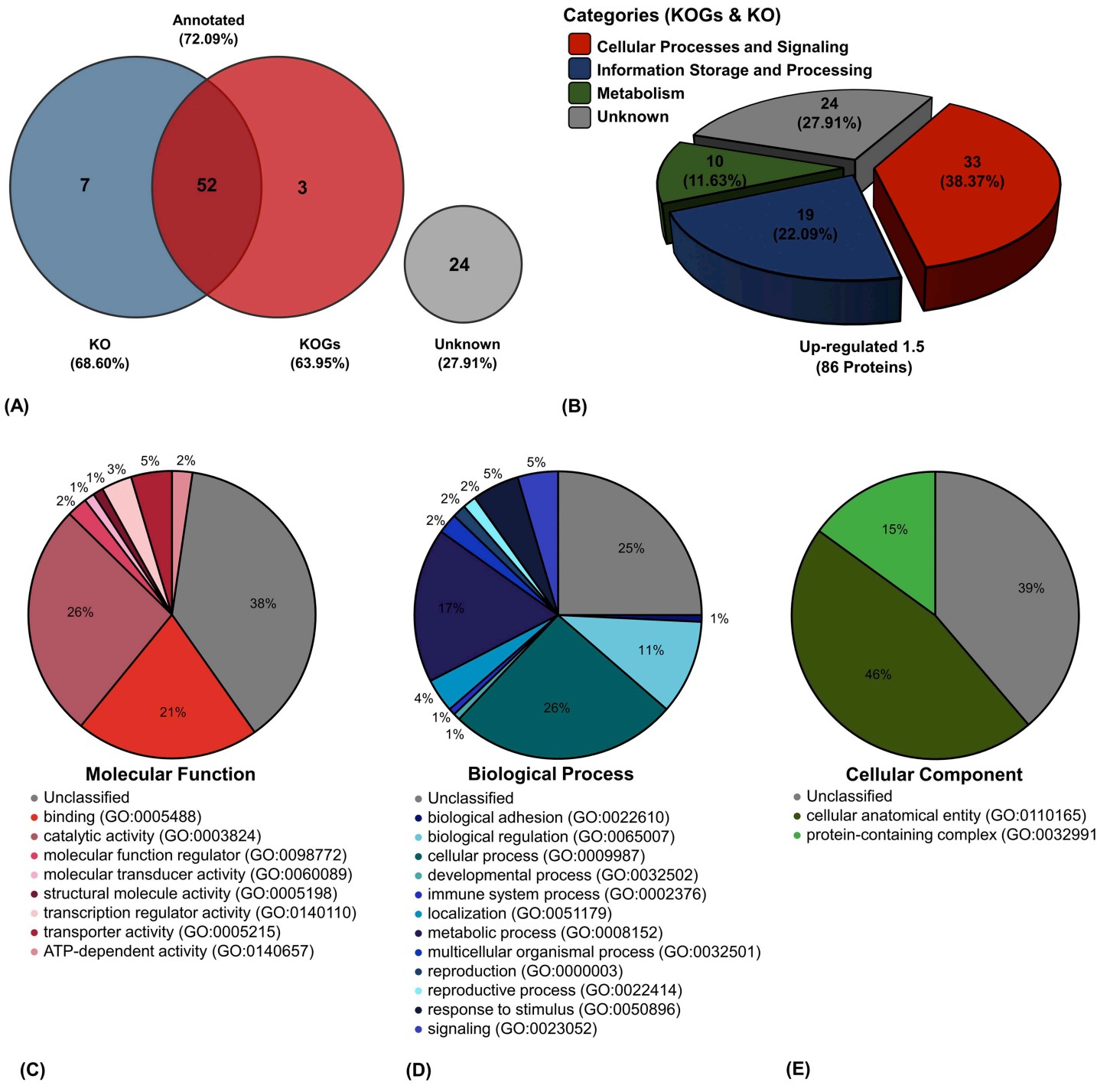

**Fig 4. Data analysis. (A)** The Venn diagram shows the comparative gene orthologous function based on KO and KOGs databases, and (B) the pie chart represents function categories when merging two databases. The gene enrichment (GO) base using PANTHER for **(C)** Molecular Function, **(D)** Biological Process, and **(E)** Cellular Component.

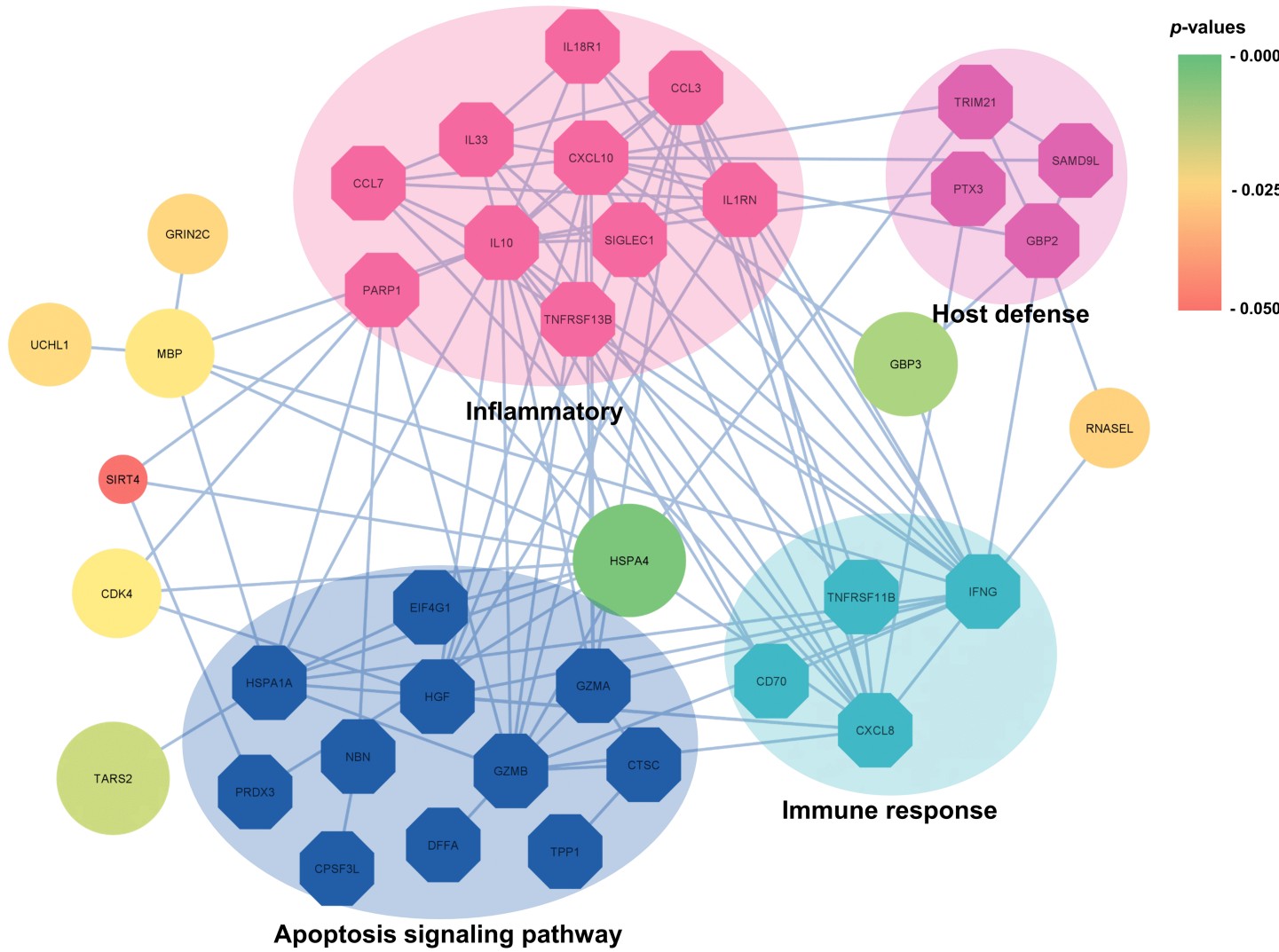

**Fig 5. Protein interaction networks of nine up-regulated reported proteins.** Octagon represents proteins associated with inflammation, host defense, the apoptosis pathway, and the immune system. The graded color of the circles represents the *p*-values of interesting proteins.

significance criteria based on statistical testing, fold-change analysis, and machine-learning evaluations. Although these proteins passed the primary screening, subsequent assessments of gene orthology and gene ontology revealed that only a subset could be functionally characterized. Ultimately, when these 86 proteins were mapped to the STRING database to construct a protein–protein interaction network, only nine exhibited meaningful or biologically relevant interaction patterns. Nine notable proteins were identified as significant in distinguishing the protein expression patterns between seropositive and naïve macaques. Within these, three were associated with the nervous system, four related to the immune system, and two were involved in metabolic processes.

Neurological symptoms associated with dengue infection have been frequently described, with previous studies identifying three types of neurological complications linked to DENV infection: metabolic alterations, viral invasion, and auto-immune responses [27]. In this study, the overexpression of GRIN2C, UCHL1, and MBP in seropositive macaques was associated with the nervous system. GRIN2C encodes a subunit of the ionotropic glutamate receptor, specifically the

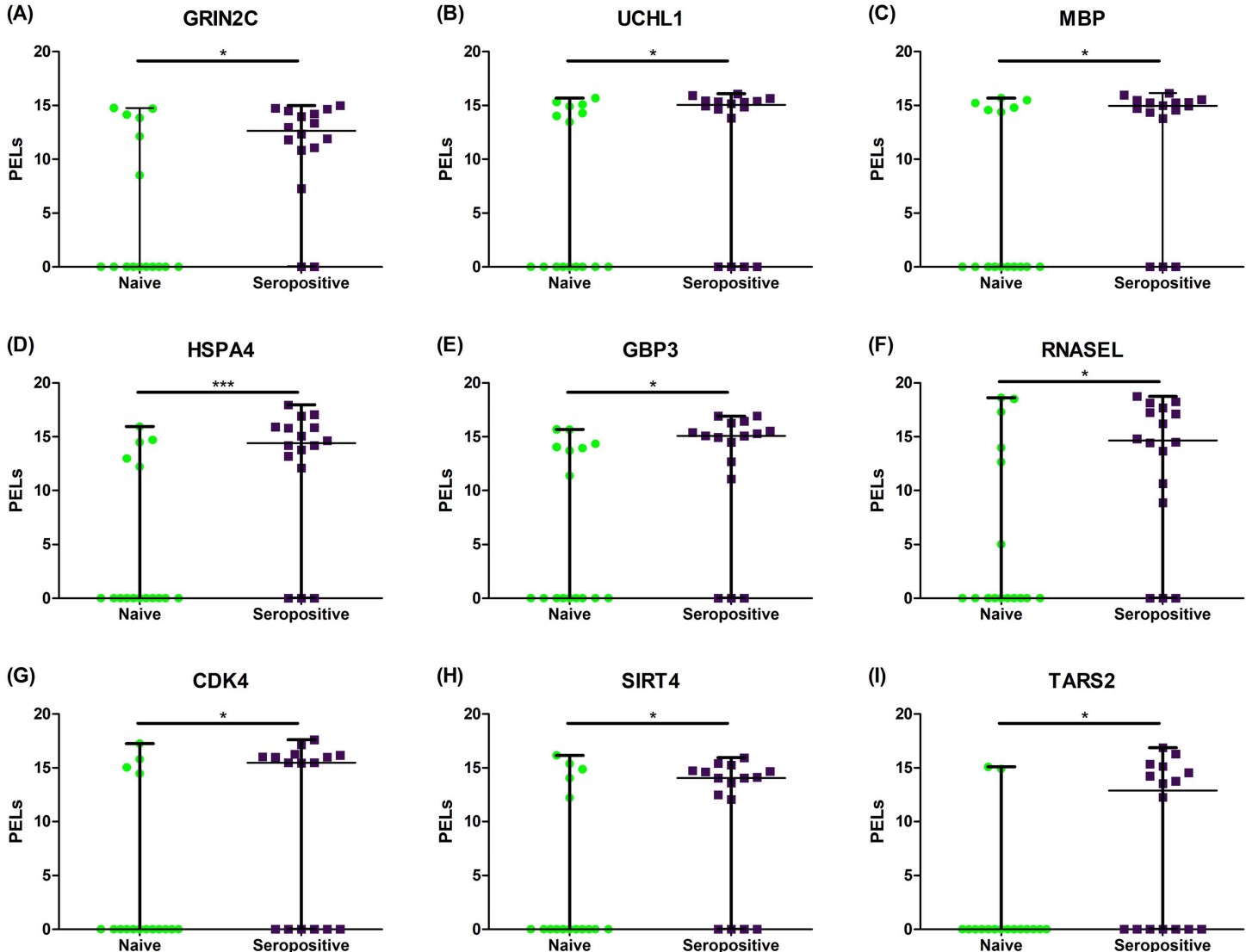

**Fig 6. The nine up-regulated interesting proteins.** The scatter dot plot represents the protein expressed level of nine up-regulated proteins between naïve (green) and DENV seropositivity (purple) namely **(A)** GRIN2C, **(B)** UCHL1, **(C)** MBP, **(D)** HSPA4, **(E)** GBP3, **(F)** RNASEL, **(G)** CDK4, **(H)** SIRT4, and **(I)** TARS2.

N-methyl-D-aspartate (NMDA) receptor [38]. NMDARs are involved in abnormal neuronal discharges, nerve conduction, inflammation, and neuronal injury. These cation-permeable receptors, expressed in the central nervous system, are critical for synaptic plasticity, memory formation, and learning [39]. The subunit composition of these receptors is implicated in neurological disorders such as Parkinson's disease, Alzheimer's disease, depression, and schizophrenia [40].

UCHL1 was initially identified as a brain-specific protein [41], but subsequent studies have shown its abundance in the fetal and adult brain as well as in the spinal cord [41]. Reduced UCHL1 activity has been linked to neurodegeneration, a hallmark of several neurological diseases. Moreover, UCHL1 is used as a neuron-derived biomarker for hemorrhagic stroke, hypoxic-ischemic encephalopathy, epilepsy, cardiac arrest, and traumatic brain injury [42]. Long HT, et al. [43] reported that gene transcript abundance patterns in the blood of children with dengue reflect differences in disease

**Table 4. 9 Up-regulated interesting proteins.**

| Initial | Protein names | Function | p-values | PELs* | | %Protein Identified | | Fold change |
|---|---|---|---|---|---|---|---|---|
| | | | | N | S | N | S | |
| GRIN2C | Glutamate ionotropic receptor NMDA type subunit 2C | Nervous system | 0.0250 | 0.00 | 12.65 | 37.50 | 87.50 | 11.64 |
| UCHL1 | Ubiquitin carboxyl-terminal hydrolase | Nervous system | 0.0240 | 0.00 | 15.05 | 43.75 | 75.00 | 14.05 |
| MBP | Myelin basic protein | Nervous system | 0.0210 | 0.00 | 14.96 | 37.50 | 81.25 | 13.96 |
| HSPA4 | Heat shock 70 kDa protein 4 | Immune system | 0.0040 | 0.00 | 14.41 | 31.25 | 81.25 | 13.41 |
| GBP3 | GB1/RHD3-type G domain-containing protein | Immune system | 0.0100 | 0.00 | 15.08 | 43.75 | 81.25 | 14.08 |
| RNASEL | Ribonuclease L | Immune system | 0.0260 | 0.00 | 14.65 | 37.50 | 81.25 | 13.65 |
| CDK4 | Cyclin-dependent kinase 4 | Immune system | 0.0200 | 0.00 | 15.50 | 25.00 | 62.50 | 14.50 |
| SIRT4 | NAD-dependent protein deacetylase sirtuin-4 | Metabolism | 0.0470 | 0.00 | 14.04 | 31.25 | 75.00 | 13.04 |
| TARS2 | Threonyl-tRNA synthetase | Metabolism | 0.0140 | 0.00 | 12.89 | 12.50 | 56.25 | 11.89 |

PELs = Protein expressed levels (Medium values), %Protein identified = (Number of Proteins identified in samples in each group/ Number of samples in each group)*100, N = Naïve group, S = Seropositive group,Fold change = [[PELs(Naïve)]-[PELs(Seropositive)]]/[PELs(Naïve)], p-values = significant with Mann–Whitney U test at p < 0.050.

progression and host responses. They found that UCHL1 was highly expressed in DSS samples collected on day four relative to the corresponding convalescent samples.

MBP (myelin basic protein) is considered a marker of active demyelination and plays a critical role in forming the myelin sheath of oligodendrocytes and Schwann cells in the CNS [44]. Glushakova OY, et al. [45]reported that MBP levels were significantly elevated at the time of hospitalization and remained high for up to two weeks in patients with severe traumatic brain injury. Elevated serum MBP levels were also associated with poorer clinical outcomes. The upregulation of MBP observed in this study suggests potential neurofunctional alterations in the seropositive wild macaques. Collectively, the upregulation of GRIN2C, UCHL1, and MBP in dengue-seropositive macaques highlights a strong association between seropositive to DENV and neurofunctional perturbations. However, it cannot be concluded that all these results are due to the direct effect of dengue; further studies are needed, as there were no clinical studies in our study. Further validation in controlled cohorts will be essential to confirm their diagnostic and pathophysiological relevance.

Additionally, four proteins identified in this study may have important functional roles. Heat shock 70 kDa protein 4 (HSPA4) is typically expressed at low levels under normal conditions but is upregulated in response to cellular stress, including specific viral infections [46]. Elevated expression of HSPA4 suggests a stress response to viral presence. As a molecular chaperone, HSPA4 is involved in protein folding, refolding misfolded proteins, and facilitating protein transport across cellular compartments. It has been identified as a potential drug target against DENV given its role in the virus-receptor complex [47]. A previous study demonstrates that inducible Hsp70 (Hsp70i) plays a crucial role in DENV pathogenesis, with its expression markedly upregulated following infection. Targeting Hsp70i with the selective allosteric inhibitor HS-72 significantly reduces viral infection by disrupting its association with the DENV receptor complex. These findings highlight Hsp70i as a viable antiviral target and support HS-72 as a promising therapeutic candidate for dengue intervention [48]. In this study, HSPA4 was upregulated and served as a hub in the protein interaction network, interacting with multiple proteins. GBPs, which are IFN-inducible and secreted by endothelial cells, contribute to antioxidant and antiviral responses. Previous research reported that markers of plasma leakage and oxidative stress were negatively correlated with GBP expression [49]. In this study, GBP3 was upregulated and connected to inflammatory and host defense pathways in the protein interaction network.

RNASEL, part of the NOD-like receptor signaling pathway, plays a key role in antiviral immunity and host defense. It degrades total RNA in coordination with *OAS* genes and *MDA5* within the IFN-I pathway, contributing to the inhibition

of RNA virus replication [50,51]. A high expression of RNASEL observed in this study may reflect an antiviral response attempting to suppress viral propagation. CDK4 is a general enzyme involved in cell cycle regulation and proliferation [51]. While its increased expression suggests heightened cellular activity, it does not necessarily indicate a direct role in dengue pathogenesis. Therefore, further investigation is required to determine its relevance to the dengue infection.

Two proteins identified in this study may have potential roles in cellular metabolism. Similar to HSPA4, SIRT4 also exhibits upregulation under cellular stress. SIRT4 belongs to the NAD+-dependent histone deacetylase family and is involved in regulating the mitochondrial function and reactive oxygen species (ROS) production [52–54]. Meuren L, et al. [55] emphasized the significance of oxidative stress as a therapeutic target in DENV infection to prevent neuroinvasion, vascular permeability, and inflammation. Threonyl-tRNA synthetase (TARS2 or TRS) functions as an aminoacyl-tRNA synthetase that catalyzes the attachment of threonine to its corresponding tRNA. Jung H, et al. [56] demonstrated that TRS promotes dendritic cell maturation and CD4+T cell polarization. They found that MAPK inhibitors reduced IκB degradation and enhanced the production of IL-12 in TRS-treated dendritic cells, indicating that MAPK pathways regulate NF-κB activity during TRS-induced immune responses. TRS has been shown to enhance dendritic cell-mediated Th1 responses and exert antiviral effects against influenza A virus through IFN-γ and TNF-α. The nine proteins identified in this study serve as potential biomarkers for DENV exposure in wild macaques, reflecting immune activation, neurofunctional changes, and metabolic stress. Their differential expression distinguishes seropositive from naïve animals and provides molecular insight into prior infection, host response, and pathways potentially linked to disease severity. While these nine proteins represent potential biomarkers of immune response in wild macaques with DENV-neutralizing antibodies, further in vitro and in vivo studies are needed to validate their functional roles.

One notable advantage of this study is its field-based design, which investigates wild macaques in their natural habitats rather than employing a controlled laboratory system. This design enables a more accurate depiction of host–virus interactions influenced by ecological and evolutionary pressures, thereby increasing the relevance of the findings to real-world zoonotic transmission. The integration of proteomic analyses with bioinformatic and pathway approaches further reinforces the reliability of the identified molecular targets. Despite these strengths, the study is constrained by a comparatively small sample size, which may limit its statistical robustness. There is also the issue of the detection of monotypic and multitypic seropositive groups that may have different effects on serum protein expression, but due to the sample size limitation, further group separation is not possible. However, the nine proteins in our study should be further studied in the future with larger sample sizes and to see trends between the monotypic and multitypic seropositive groups.

## Conclusions

Our study highlights emerging challenges in the investigation of serum proteins in wild macaques with neutralizing antibodies against DENV. Nine proteins were identified with significantly upregulated expression in the seropositive group, distinguishing them from naïve individuals. Among these, three proteins were associated with the nervous system, while six had potential relevance to anti-DENV activity. HSPA4 is frequently linked to cellular stress and has been proposed as a potential antiviral target. GBP3 indicates a heightened host immune response via IFN signaling, and RNASEL, commonly involved in RNA virus clearance, was highly expressed. CDK4 suggests increased cellular activity, while SIRT4 is upregulated under cellular stress. TARS2, a general aminoacyl-tRNA synthetase, also contributes to dendritic cell activation and Th1-mediated immune responses through IFN-γ and TNF-α. Although the antiviral potential of these six proteins is to be confirmed, their elevated levels in seropositive macaques, alongside increased expression of three nervous system proteins, suggest their relevance in post-infection responses. Although these proteins have been associated with the nervous system and have been implicated in anti-DENV responses, the macaques in this study were only previously infected. Nonetheless, these proteins remain up-regulated post-infection, warranting further investigation into their dynamics during long-term post-infection periods. While human serum proteomic profiling in dengue is well-studied, corresponding

analyses in non-human primates remain limited and warrant further investigation for their value in understanding immune cross-reactivity.

## Acknowledgments

The authors would like to thank the staff of the Department of National Parks, Wildlife and Plant Conservation, Thailand for their assistance in trapping the stump-tailed macaques.

## Author contributions

**Conceptualization:** Wirasak Fungfuang.

**Data curation:** Wirasak Fungfuang, Pakorn Ruengket, Sittiruk Roytrakul.

**Funding acquisition:** Wirasak Fungfuang.

**Methodology:** Wirasak Fungfuang, Pakorn Ruengket, Sittiruk Roytrakul, Daraka Tongthainan, Kobporn Boonnak, Kanokwan Taruyanon, Bencharong Sangkharak.

**Resources:** Daraka Tongthainan.

**Software:** Sittiruk Roytrakul.

**Writing – original draft:** Pakorn Ruengket.

**Writing – review & editing:** Wirasak Fungfuang, Pakorn Ruengket, Sittiruk Roytrakul, Daraka Tongthainan, Kobporn Boonnak, Kanokwan Taruyanon, Bencharong Sangkharak.

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
