## [Decision Letter · Decision Letter 0]

17 Nov 2025

Dear Dr. Fungfuang,

Thank you for submitting your manuscript to PLOS ONE. After careful consideration, we feel that it has merit but does not fully meet PLOS ONE’s publication criteria as it currently stands. Therefore, we invite you to submit a revised version of the manuscript that addresses the points raised during the review process.

We look forward to receiving your revised manuscript.

Kind regards,

Harapan Harapan, MD, PhD

Academic Editor

PLOS ONE

“Thailand Research Fund, Thailand (Grant No. MRG 6080260).”

“W.F. was in receipt of grants from the Thailand Research Fund, Thailand (Grant No. MRG 6080260). The funder had no role in the design and conduct of the study; collection, management, analysis, and interpretation of the data; preparation, review, or approval of the manuscript; and the decision to submit the manuscript for publication.”

“Thailand Research Fund, Thailand (Grant No. MRG 6080260).”

Reviewers' comments:

Reviewer's Responses to Questions

**Comments to the Author**

1. Is the manuscript technically sound, and do the data support the conclusions?

Reviewer #1: Partly

Reviewer #2: Yes

Reviewer #3: Partly

2. Has the statistical analysis been performed appropriately and rigorously?

Reviewer #1: Yes

Reviewer #2: Yes

Reviewer #3: Yes

3. Have the authors made all data underlying the findings in their manuscript fully available?

Reviewer #1: No

Reviewer #2: Yes

Reviewer #3: Yes

4. Is the manuscript presented in an intelligible fashion and written in standard English?

Reviewer #1: Yes

Reviewer #2: Yes

Reviewer #3: Yes

Reviewer #1: This manuscript explores the serum proteomic characterization of wild stump-tailed macaques with and without dengue virus neutralizing antibodies in Thailand. The study identifies differentially expressed proteins (DEPs) potentially linked to antiviral responses and neurodegenerative processes. The findings may provide preliminary insights into host responses in natural dengue reservoirs. However, there are several major comments that need to be addressed:

Major Comments:

1. Infection with flaviviruses can induce cross-reactive antibodies. The authors need to provide PRNT90 data for other flaviviruses, such as the Zika virus, to confirm that the serum samples are specific to DENV and not influenced by cross-reactivity. The manuscript should clearly state the criteria used to rule out cross-reactivity from other flavivirus infections.

2. The manuscript does not fully explain the rationale for examining serum proteomic profiles in monkeys. Please clarify the benefits of this study and how it contributes to the understanding of the disease. This rationale should be clearly stated in the introduction and reinforced in the conclusion.

3. The differential expression analysis was performed to compare between naive and seropositive monkeys. However, it is a significant limitation that the seropositive group was not further stratified. Could the authors analyze whether there are any differential expressions between monotypic and multitypic seropositive monkeys? The distinct immune responses to monotypic vs multitypic dengue infections might influence host protein profiles.

4. The phrase “sequelae of dengue infection” is used, but the macaques were not clinically characterized. Without longitudinal or clinical follow-up, it is difficult to conclude on sequelae. The wording should be revised to accurately reflect this limitation.

5. The use of 1.2- and 1.5-fold thresholds appears arbitrary. Please justify the selection of these specific thresholds. Please clarify why the 1.5-fold threshold was not used for the initial analysis.

6. The criteria for selecting the nine interesting proteins expressed in seropositive monkeys are unclear. The author states that 582 DEPs were identified, with 560 of these shared between the naive and seropositive groups. Please give a clear explanation of the

“shared between” proteins. Additionally, the rationale for focusing on the nine proteins after filtered methods needs to be explained.

7. The author focuses mainly on upregulated proteins. Downregulated proteins and those uniquely DEPs expressed in naïve macaques also need to be analyzed and discussed.

8. While the discussion provides extensive descriptions of the nine proteins, some interpretations seem speculative. For example, the link between upregulated proteins and neurological complications in macaques is overstated, as no clinical data of macaques were collected. The text should be revised to focus on observed associations.

9. In Figure 6, it is unclear whether the PELs correspond to the results shown in Table 4, especially since naïve samples are shown as 0. The bar graph in Figure 6 does not clearly show whether a significant difference exists between the naive and seropositive samples

Minor Comments:

1. Please add the reference “Additionally, the so-called sylvatic isolate of DENV-1 is not phylogenetically distinct from human-derived lineages, leading to uncertainty about its origins”

2. Please provide specific details about the DENV strains used in the study.

3. Line 262: “Affinity propagation” should be briefly explained.

4. Line 273 The “three filter-based methods” should be briefly stated to clarify the methods applied.

5. The statistical data, such as fold-change values, should be added to Table 4

6. Ensure consistent formatting of gene/protein names (e.g., CDK4, TARS2).

7. Please ensure in how research teams are referenced throughout the manuscript, following conventional citation styles, for example, the format 'Tongthainan, Mongkol (21)' should be revised to a conventional style such as 'Tongthainan D, et al.

8. Line 377 “HS-72” should be briefly explained.

9. Figures should be provided in higher resolution.

10. I cannot access the data deposited from the link https://repository.jpostdb.org/preview/851514434663b785894b48. Please check the access code.

Reviewer #2: Reviewer Comments

Overall

The human serum proteomic profiling in dengue is well-studied, corresponding analyses in non-human primates remain limited. This manuscript investigates the serum proteomic profiles of wild macaques with (seropositive) and without (naive) dengue virus-neutralizing antibodies. However, the grouping of wild macaques with seropositive sera for testing is not presented.

The following points may improve the manuscript:

Introduction

Could you please provide more information on the DENV infections in non-human primates, in term of disease manifestation (Symptomatic/Asymptomatic), ADE, viral transmission?

Methods

Could you please describe more on the locations of baited cages or provide the map, and sex and estimated age of all macaques in the study?

In serum protein preparation section, did you use each serum sample (individual macaque) or the pooled of serum sample (naïve, and seropositive groups)?

Results

In Neutralizing antibody against DENV section, could you please describe more on the sex (confounding factors in proteomic profiles) and estimated age (naïve/monotypic/multipletypic may relate to age) in the study?

In Differential protein identification between two macaque groups, could you please classify more on the naïve/monotypic/multipletypic and the low/moderate/high NT subgroup analysis?

Discussion

Could you please explain why these nine proteins represent potential biomarkers in wild macaques for any aspect (transmission, or severity)?

Reviewer #3: 1. (Introduction): Please strengthen the explanation of the immune response against the dengue virus. Please check and cite the following article: https://doi.org/10.52225/narra.v4i1.309.

2. Please explicitly explain the rationale for choosing a sample size of 32 macaques. Is this sample size sufficient, or is it relatively small? If it is indeed relatively small, you must acknowledge this as a limitation and suggest that these findings should be validated with a larger sample size in the future.

3. Please perform a thorough proofread of the entire manuscript to avoid typos such as "mornitor," "temporaty," and others.

4. Please check the manuscript for the correct formatting of 'p-value'; it should be written in lowercase, not with a capital 'P'.

5. (Materials and Methods, Line 138): Please provide a stronger explanation for why the PRNT90 ≥20 criterion was used as the threshold for seropositivity.

6. Consider adding a table containing the characteristics of the two study groups, such as estimated age, sex, and physical condition. This table is necessary to ensure that the two sample groups are comparable.

7. Please explain explicitly the process used to select only 9 proteins for further analysis from the total of 86 upregulated proteins.

8. The statement that GRIN2C, UCHL1, and MBP proteins indicate neurodegenerative damage is too strong and speculative without direct supporting evidence. This interpretation should be presented as a hypothesis, not a conclusion, as these proteins can originate from various sources (such as peripheral nerves or immune cells).

**Do you want your identity to be public for this peer review?** For information about this choice, including consent withdrawal, please see our Privacy Policy

Reviewer #1: No

Reviewer #2: No

Reviewer #3: No

---

## [Author Response · Author response to Decision Letter 1]

9 Dec 2025

Response to reviewers

Reviewer #1: This manuscript explores the serum proteomic characterization of wild stump-tailed macaques with and without dengue virus neutralizing antibodies in Thailand. The study identifies differentially expressed proteins (DEPs) potentially linked to antiviral responses and neurodegenerative processes. The findings may provide preliminary insights into host responses in natural dengue reservoirs. However, there are several major comments that need to be addressed:

Major Comments:

1. Infection with flaviviruses can induce cross-reactive antibodies. The authors need to provide PRNT90 data for other flaviviruses, such as the Zika virus, to confirm that the serum samples are specific to DENV and not influenced by cross-reactivity. The manuscript should clearly state the criteria used to rule out cross-reactivity from other flavivirus infections.

Response: Thank you for your concern and nice recommendation. We concur with the reviewers that cross-reactivity among flaviviruses is evident, particularly when assessing neutralizing antibodies between dengue and Zika viruses. Consequently, to distinguish ZIKV infection from other flavivirus infections using PRNT90 antibody titers, we interpreted the PRNT90 test results according to WHO criteria (PMID: 29943723, PMID: 26897760, PMID: 32588813). According to these criteria, ZIKV infection is classified in samples with PRNT90 titer values greater than 20 and a 4-fold difference between ZIKV and DENV PRNT90 titers. We have incorporated this information to clarify the cross-reactivity issues in the revised manuscript, specifically in lines 174-177.

2. The manuscript does not fully explain the rationale for examining serum proteomic profiles in monkeys. Please clarify the benefits of this study and how it contributes to the understanding of the disease. This rationale should be clearly stated in the introduction and reinforced in the conclusion.

Response: Thank you for your review and lovely comments. We have revised the manuscript on your suggestion in the Abstract, Introduction, Discussion, and Conclusion (Red letters).

3. The differential expression analysis was performed to compare between naive and seropositive monkeys. However, it is a significant limitation that the seropositive group was not further stratified. Could the authors analyze whether there are any differential expressions between monotypic and multitypic seropositive monkeys? The distinct immune responses to monotypic vs multitypic dengue infections might influence host protein profiles.

Response: Thank you for your advice. The issue of separating monotypic and multitypic seropositive groups is an interesting and important point. However, our study has limitations in sample size and homogeneity of each group that may affect the statistical reliability, so we cannot analyze the groups separately. However, we have added these limitations at the end of the discussion section as per your suggestion in lines 467-478.

4. The phrase "sequelae of dengue infection" is used, but the macaques were not clinically characterized. Without longitudinal or clinical follow-up, it is difficult to conclude on sequelae. The wording should be revised to accurately reflect this limitation.

Response: Thank you for your valuable advice. We have adjusted the wording to reduce the chance of misunderstanding in line 49.

5. The use of 1.2- and 1.5-fold thresholds appears arbitrary. Please justify the selection of these specific thresholds. Please clarify why the 1.5-fold threshold was not used for the initial analysis.

Response: We appreciate the reviewer’s comment regarding the choice of fold-change thresholds. Selection criteria are inherently flexible and should be aligned with the specific objectives of the study. In this work, we aimed to generate a serum proteomic profile to identify potential biomarker candidates. Thresholds set too low would yield an excessively large number of proteins, whereas overly stringent thresholds would substantially limit the dataset. Accordingly, we evaluated both 1.2- and 1.5-fold cutoffs to balance sensitivity and stringency, ensuring that neither an overabundance nor an insufficiency of proteins was included.

6. The criteria for selecting the nine interesting proteins expressed in seropositive monkeys are unclear. The author states that 582 DEPs were identified, with 560 of these shared between the naive and seropositive groups. Please give a clear explanation of the "shared between" proteins. Additionally, the rationale for focusing on the nine proteins after filtering needs to be explained.

Response: Thanks for the suggestion. We have added information about using the filter-based method in the discussion section to highlight its importance in lines 377-387.

7. The author focuses mainly on upregulated proteins. Downregulated proteins and those uniquely DEPs expressed in naïve macaques also need to be analyzed and discussed.

Response: Thank you for your review and lovely comments. The objective of this study was to identify candidate proteins suitable for development as serum biomarkers. Our analysis up-regulated proteins whose abundance increased in response to infection. Serum proteins originate from tissues or cells that actively secrete proteins or release them following cellular damage; consequently, they cannot be readily attributed to specific cell or tissue sources. Prioritizing up-regulated proteins enhances their potential detectability and applicability in diagnostic assays. Although the investigation of down-regulated proteins is scientifically valuable, their translational utility as biomarkers is comparatively limited, making them less practical for further development as biomarkers in lines 377-387.

8. While the discussion provides extensive descriptions of the nine proteins, some interpretations seem speculative. For example, the link between upregulated proteins and neurological complications in macaques is overstated, as no clinical data of macaques were collected. The text should be revised to focus on observed associations.

Response: Thank you for your lovely and valuable advice. We have been more careful with our wording and emphasized the limitations of our study in the abstract, discussion, and conclusion.

9. In Figure 6, it is unclear whether the PELs correspond to the results shown in Table 4, especially since naïve samples are shown as 0. The bar graph in Figure 6 does not clearly show whether a significant difference exists between the naive and seropositive samples

Response: Thank you for your lovely comments and reminders. From Table 4, we have presented the PELs values with the median value, since the statistic we used is the Mann-Whitney U test, which uses the median value in statistical analysis, while the graph in Figure 6 shows the median value at zero; however, we have edited the image to make it clearer with a scatter dot plot.

Minor Comments:

1. Please add the reference "Additionally, the so-called sylvatic isolate of DENV-1 is not phylogenetically distinct from human-derived lineages, leading to uncertainty about its origins"

Response: Thank you for your review and lovely comments. We have revised the manuscript on your suggestion in lines 85-87.

2. Please provide specific details about the DENV strains used in the study.

Response: Thank you for your strict and kind review. We have listed the dengue strain information in Table 2.

3. Line 262: "Affinity propagation" should be briefly explained.

Response: Thank you for your review and lovely comments. We have revised the manuscript on your suggestion.

4. Line 273 The "three filter-based methods" should be briefly stated to clarify the methods applied.

Response: Thanks for the suggestion. We have added information about using the filter-based method in the discussion section to highlight its importance in lines 377-387.

5. The statistical data, such as fold-change values, should be added to Table 4

Response: Thank you for your constructive comments. We have incorporated the additional information in accordance with your recommendations in Table 4.

6. Ensure consistent formatting of gene/protein names (e.g., CDK4, TARS2).

Response: Thank you for your lovely comments and reminders. We have rechecked and confirmed.

7. Please ensure that in how research teams are referenced throughout the manuscript, following conventional citation styles, for example, the format 'Tongthainan, Mongkol (21)' should be revised to a conventional style such as 'Tongthainan D, et al.

Response: Thank you for your review and lovely comments. We have revised the manuscript on your suggestion in line 117.

8. Line 377 "HS-72" should be briefly explained.

Response: Thank you for your review and lovely comments. We have revised the manuscript on your suggestion in line 429-434.

9. Figures should be provided in higher resolution.

Response: Thank you for your review and lovely comments. We have revised the manuscript on your suggestion.

10. I cannot access the data deposited from the link

https://repository.jpostdb.org/preview/851514434663b785894b48. Please check the access code.

Response: Thank you for your review and lovely comments. We have revised the manuscript on your suggestion in PXD052111(https://proteomecentral.proteomexchange.org/cgi/GetDataset?ID=PXD052111).

Reviewer #2: Reviewer Comments

Overall

The human serum proteomic profiling in dengue is well-studied, corresponding analyses in non-human primates remain limited. This manuscript investigates the serum proteomic profiles of wild macaques with (seropositive) and without (naive) dengue virus-neutralizing antibodies. However, the grouping of wild macaques with seropositive sera for testing is not presented.

The following points may improve the manuscript:

Introduction

Could you please provide more information on the DENV infections in non-human primates, in term of disease manifestation (Symptomatic/Asymptomatic), ADE, viral transmission?

Response: Thank you for your review and lovely comments. We have revised the manuscript on your suggestion in lines 95-108.

Methods

Could you please describe more on the locations of baited cages or provide the map, and sex and estimated age of all macaques in the study? In serum protein preparation section, did you use each serum sample (individual macaque) or the pooled of serum sample (naïve, and seropositive groups)?

Response: Thank you for your valuable feedback. We have added the GPS data, sex, and age of the macaques in the methods section as per your suggestion in lines 141-143. For the protein analysis section, we used the serum proteins of each macaque for the analysis.

Results

In the Neutralizing antibody against DENV section, could you please describe more on the sex (confounding factors in proteomic profiles) and estimated age (naïve/monotypic/multipletypic may relate to age) in the study?

In Differential protein identification between two macaque groups, could you please classify more on the naïve/monotypic/multipletypic and the low/moderate/high NT subgroup analysis?

Response: Thank you for your advice. We have added the GPS data, sex, and age of the macaques in the methods section as per your suggestion in lines 141-143. For the protein analysis section, we used the serum proteins of each macaque for the analysis. The issue of separating monotypic and multitypic seropositive groups is an interesting and important point. However, our study has limitations in sample size and homogeneity of each group that may affect the statistical reliability, so we cannot analyze the groups separately. However, we have added these limitations at the end of the discussion section as per your suggestion in lines 467-478.

Discussion

Could you please explain why these nine proteins represent potential biomarkers in wild macaques for any aspect (transmission, or severity)?

Response: Thank you for your review and lovely comments. We have revised the manuscript on your suggestion in lines 460-464.

Reviewer #3:

1. (Introduction): Please strengthen the explanation of the immune response against the dengue virus. Please check and cite the following article: https://doi.org/10.52225/narra.v4i1.309.

Response: Thank you for your concern and nice recommendation. We have edited according to your suggestion in line 66-68.

2. Please explicitly explain the rationale for choosing a sample size of 32 macaques. Is this sample size sufficient, or is it relatively small? If it is indeed relatively small, you must acknowledge this as a limitation and suggest that these findings should be validated with a larger sample size in the future.

Response: Thank you for your nice comment. We also noted the limitations of the small sample size in the discussion in lines 467-478.

3. Please perform a thorough proofread of the entire manuscript to avoid typos such as "mornitor," "temporaty," and others.

Response: Thank you for your concern and nice recommendation. We have edited according to your suggestion in lines 145 and 151.

4. Please check the manuscript for the correct formatting of 'p-value'; it should be written in lowercase, not with a capital 'P'.

Response: Thank you for your concern and nice recommendation. We have edited according to your suggestion.

5. (Materials and Methods, Line 138): Please provide a stronger explanation for why the PRNT90 ≥20 criterion was used as the threshold for seropositivity.

Response: Thank you for your concern and nice recommendation. We applied WHO criteria to determine DENV seropositivity, considering neutralizing antibody titers of 1:20 or higher as positive ( PMID: 29943723, PMID: 26897760, PMID: 32588813). We have incorporated this information, along with the cited reference, into the Methods section of the revised manuscript (Lines 174-177).

6. Consider adding a table containing the characteristics of the two study groups, such as estimated age, sex, and physical condition. This table is necessary to ensure that the two sample groups are comparable.

Response: Thank you for your valuable feedback. We have added the GPS data, sex, and age of the macaques in the methods section as per your suggestion in lines 141-143. For the protein analysis section, we used the serum proteins of each macaque for the analysis.

7. Please explain explicitly the process used to select only 9 proteins for further analysis from the total of 86 upregulated proteins.

Response: Thanks for the suggestion. We have added information about using the filter-based method in the discussion section to highlight its importance in lines 377-387.

8. The statement that GRIN2C, UCHL1, and MBP proteins indicate neurodegenerative damage is too strong and speculative without direct supporting evidence. This interpretation should be presented as a hypothesis, not a conclusion, as these proteins can originate from various sources (such as peripheral nerves or immune cells).

Response: Thank you for your lovely and valuable advice. We have been more careful with our wording and emphasized the limitations of our study in the abstract, discussion, and conclusion.

---

## [Decision Letter · Decision Letter 1]

29 Dec 2025

Dear Dr. Fungfuang,

Thank you for submitting your manuscript to PLOS ONE. After careful consideration, we feel that it has merit but does not fully meet PLOS ONE’s publication criteria as it currently stands. Therefore, we invite you to submit a revised version of the manuscript that addresses the points raised during the review process.

We look forward to receiving your revised manuscript.

Kind regards,

Harapan Harapan, MD, PhD

Academic Editor

PLOS One

Journal Requirements:

Reviewers' comments:

Reviewer's Responses to Questions

**Comments to the Author**

Reviewer #1: (No Response)

Reviewer #2: All comments have been addressed

2. Is the manuscript technically sound, and do the data support the conclusions?

Reviewer #1: Yes

Reviewer #2: Yes

3. Has the statistical analysis been performed appropriately and rigorously?

Reviewer #1: Yes

Reviewer #2: Yes

4. Have the authors made all data underlying the findings in their manuscript fully available?

Reviewer #1: Yes

Reviewer #2: Yes

5. Is the manuscript presented in an intelligible fashion and written in standard English?

Reviewer #1: Yes

Reviewer #2: Yes

Reviewer #1: Overall, The revision is mostly satisfactory, but there are a few minor points that should be addressed.

1. Please check and correct Table 2, as the row for DENV-1/DENV-3 is repeated.

2. Please ensure consistent formatting of gene/protein names throughout the manuscript (e.g., Line 453 refers to Threonyl-tRNA synthetase (TRS), whereas Table 4 uses TARS2; Line 488 uses CDK instead of CDK4)

3. Please ensure in how research teams are referenced throughout the manuscript, following conventional citation styles (Line 451Meuren, Prestes, and Line 455,Jung Park)

Reviewer #2: Serum proteomic characterization of Stump-tailed Macaques (Macaca arctoides) with neutralizing antibodies against Dengue virus in Thailand

Reviewer Comments

This manuscript investigates the serum proteomic profiles of wild macaques with (seropositive) and without (naive) dengue virus-neutralizing antibodies. Unfortunately, the grouping of wild macaques with seropositive sera for testing is not presented.

**Do you want your identity to be public for this peer review?** For information about this choice, including consent withdrawal, please see our Privacy Policy

Reviewer #1: No

Reviewer #2: No

---

## [Author Response · Author response to Decision Letter 2]

2 Jan 2026

Dear Editor

We would like to express our appreciation for your assistance and the reviewers' comments on our manuscript titled “Serum proteomic characterization of Stump-tailed Macaques (Macaca arctoides) with neutralizing antibodies against Dengue virus in Thailand” (PONE-D-25-34021R1). All opinions are valuable and will help us revise and improve our manuscript. We carefully reviewed all comments, made adjustments, and highlighted the changed section in red in the manuscript. We are hopeful that our updated text will be satisfactory and approved.

Reviewer #1:

Overall, the revision is mostly satisfactory, but there are a few minor points that should be addressed.

1. Please check and correct Table 2, as the row for DENV-1/DENV-3 is repeated.

Response: Thank you for your strict review and lovely comments. We have revised the manuscript on your suggestion.

2. Please ensure consistent formatting of gene/protein names throughout the manuscript (e.g., Line 453 refers to Threonyl-tRNA synthetase (TRS), whereas Table 4 uses TARS2; Line 488 uses CDK instead of CDK4)

Response: Thank you for your strict review and lovely comments. We have revised the manuscript on your suggestion.

3. Please ensure in how research teams are referenced throughout the manuscript, following conventional citation styles (Line 451Meuren, Prestes, and Line 455, Jung Park)

Response: Thank you for your strict review and lovely comments. We have revised all references in the manuscript on your suggestion.

Reviewer #2:

Serum proteomic characterization of Stump-tailed Macaques (Macaca arctoides) with neutralizing antibodies against Dengue virus in Thailand

Reviewer Comments

This manuscript investigates the serum proteomic profiles of wild macaques with (seropositive) and without (naive) dengue virus-neutralizing antibodies. Unfortunately, the grouping of wild macaques with seropositive sera for testing is not presented.

Response: Thank you for your lovely and valuable advice.

Thank you for your consideration of this manuscript.

---

## [Editor Report · Decision Letter 2]

6 Jan 2026

Serum proteomic characterization of Stump-tailed Macaques (Macaca arctoides) with neutralizing antibodies against Dengue virus in Thailand

PONE-D-25-34021R2

Dear Dr. Fungfuang,

We’re pleased to inform you that your manuscript has been judged scientifically suitable for publication and will be formally accepted for publication once it meets all outstanding technical requirements.

Kind regards,

Harapan Harapan, MD, PhD

Academic Editor

PLOS One
---

## [Editor Report · Acceptance letter]

PONE-D-25-34021R2

PLOS One

Dear Dr. Fungfuang,

I'm pleased to inform you that your manuscript has been deemed suitable for publication in PLOS One. Congratulations! Your manuscript is now being handed over to our production team.

Kind regards,

on behalf of

Dr. Harapan Harapan

Academic Editor

PLOS One